# A darkening spring: How preexisting distrust shaped COVID-19 skepticism

**J. Hunter Priniski**[1]*, **Keith J. Holyoak**[1,2]

**1** Department of Psychology, University of California, Los Angeles, CA, United States of America, **2** Brain Research Institute, University of California, Los Angeles, CA, United States of America

* priniski@ucla.edu

**Data Availability Statement:** Survey data and R scripts for reproducing the reported analyses and figures are available open access at the Open Science Framework: https://osf.io/8xerq/?view_only=df620c3e86984668ae1a225028b5cd9b.

## Abstract

Despite widespread communication of the health risks associated with the COVID-19 virus, many Americans underestimated its risks and were antagonistic regarding preventative measures. Political partisanship has been linked to diverging attitudes towards the virus, but the cognitive processes underlying this divergence remain unclear. Bayesian models fit to data gathered through two preregistered online surveys, administered before (March 13, 2020, $N$ = 850) and during the first wave (April-May, 2020, $N$ = 1610) of cases in the United States, reveal two preexisting forms of distrust—distrust in Democratic politicians and in medical scientists—that drove initial skepticism about the virus. During the first wave of cases, additional factors came into play, suggesting that skeptical attitudes became more deeply embedded within a complex network of auxiliary beliefs. These findings highlight how mechanisms that enhance cognitive coherence can drive anti-science attitudes.

## Significance

The COVID-19 pandemic highlighted stark divisions in trust of public-health recommendations made by medical scientists and government officials. We conducted surveys of beliefs about COVID-19 severity and related attitudes in the spring of 2020, just prior to and then during the first wave of COVID-19 cases in the United States. Two preexisting varieties of distrust––in Democratic politicians and in medical scientists––shaped early skepticism about the threat posed by COVID-19. Even before the disease had impacted the U.S., fear of a potential COVID-19 vaccine (including the possibility of mandated vaccination) already fostered skepticism about COVID-19 severity. By the time of the first wave, the initial seeds of skepticism had broadened to include additional factors, such as distrusting large medical organizations. These findings highlight how preexisting distrust of expert sources of information can enable political rhetoric and active misinformation campaigns to bolster anti-science attitudes and thereby undermine support for prosocial action.

## Introduction

From fighting climate change to curbing the spread of an infectious disease, collective efforts are necessary to overcome threats facing society [1]. When relatively rare expertise is required

**Funding:** Preparation of this paper was supported by NSF Grant BCS-1827374. The funders of this study had no role in study design, data collection and analysis, decision to publish, or preparation of the manuscript.

**Competing interests:** The authors have declared that no competing interests exist.

to directly obtain relevant evidence, then laypeople's beliefs and behaviors are likely to depend on acceptance of scientific information provided by experts. However, when domain experts suggest that people change their behaviors, many laypeople distrust the veracity of these sources and continue making choices contrary to what the available evidence supports [2]. To effectively communicate information that inspires prosocial behaviors, it is necessary to understand the cognitive processing leading to rejection of expert recommendations.

The need for science-backed decision making was evident during the COVID-19 pandemic. As of summer 2021, the pandemic had taken over 600,000 American lives [3] and cost the country an estimated $16 trillion [4]. Nonetheless, a large proportion of Americans remained skeptical of the severity of the virus [5]. Even in their final breaths, some who died from the illness refused to acknowledge its existence [6]. How can people's beliefs be so resistant to disconfirming evidence and direct experience? What prevents scientific information from having its intended impact on laypeople's behavior?

A large body of work suggests that political ideology predicts people's attitudes about COVID-19 and endorsement of related misconceptions. In the U.S. and elsewhere, liberals were dramatically more likely than conservatives to recognize the severe health risks associated with contracting the virus [7,8]. This difference shaped willingness to reduce its spread by social distancing [9,10], wearing a mask [11], and receiving a COVID-19 vaccine [12]. These divergences are representative of more general trends concerning how Democrats and Republicans hold opposing views about science issues––most notably, polarized beliefs about the reality of climate change [13]. The core beliefs associated with political ideologies are presumably political rather than scientific, and it is likely some general cognitive mechanism connects these two belief systems so that political worldviews impact beliefs about science [14,15].

At a fundamental level, people rarely hold beliefs in isolation. Rather, people hold systems of beliefs that mutually *cohere* [16–18]. Following a long tradition in Gestalt psychology [19], elements in a network of beliefs, attitudes, or emotional responses [20] are connected by positive or negative links based on causal or inferential relations, associations, or emotional valence. To maximize coherence, people tend to adjust their beliefs in a way that increases their consistency. In some cases, adjustments to maximize coherence may have a rational basis, given people's prior beliefs [21,22], but in other cases coherence may simply reflect motivated reasoning [23]—a tendency to believe what one hopes for. For example, people who strongly support free market policies, which they consider to be threatened by efforts to prevent anthropogenic climate change, may "protect" their political beliefs by accepting allegations that climate scientists are biased towards producing evidence for climate change [15,24], so that scientific evidence may have no impact on beliefs [25]. More generally, beliefs about science will tend to cohere with other beliefs (e.g., about politics and morality). Misconceptions about science and medicine may stem from the motivation to preserve core political ideology, and to maintain ties with certain social and political groups [26]. The tendency to maintain coherence may impact how (or whether) beliefs are revised when considering scientific data.

It follows that doubts about the credibility of established sources of expert knowledge (e.g., academic scientists, journalists, medical organizations) can result in failures to revise beliefs despite exposure to evidence. Political rhetoric and misinformation can serve to create and maintain doubt about the credibility of expert sources [27–29]. In particular, Republican politicians often ignored the threat posed by the coronavirus [8], and actively sowed public distrust in medical expertise [30]. For example, President Trump publicly insisted over forty times that the virus will "like a miracle—disappear" [31], and his former Chief-of-Staff Steven Bannon called for the beheading of Dr. Anthony Fauci [32], the country's leading infectious disease expert. This type of rhetoric has a causal impact on anti-science attitudes among conservatives

[33]. Online, the COVID-19 pandemic was accompanied by an "infodemic" with medical misinformation and conspiracy theories spreading [34]. For instance, during the summer of 2020, the viral conspiracy video *Plandemic*: *The Hidden Agenda Behind COVID-19* claimed scientists planned the COVID-19 pandemic for financial profit and to strip Americans of their liberties [35]. Months before a vaccine was even developed, conspiracy theories were already spreading (e.g., alleging that Bill Gates was using the COVID-19 vaccine to plant tracking devices in people's bodies) [36].

In the present study we sought to identify prior beliefs that allowed political rhetoric and misinformation about COVID-19 to take root and shape people's attitudes. We report a study of interrelated beliefs held by Americans just before the first wave of cases struck the United States (in March 2020), and then amid the first wave (April-May 2020). By examining patterns of beliefs and attitudes at the very beginning of the pandemic in the United States, we may be able to detect the seeds of the partisan divide that would emerge over the later course of the disease and its aftermath and illuminate the mechanisms by which rhetoric misinformation took hold.

## Method

### Ethics statement

The Institutional Review Board of the University of California, Los Angeles, approved the study, and written consent was received for each participant. All participants were over 18 years of age.

### Participants

We administered surveys on three separate days, each approximately a month apart (see Fig 1). The *prewave* survey, administered on March 13, 2020, involved 831 participants recruited from Amazon's Mechanical Turk (MTurk). We analyzed 628 subjects who passed seven attention checks placed throughout the survey. The *first-wave* survey (which combined the second and third administrations) involved 1700 participants recruited from MTurk (850 on April 17, 850 on May 19). We analyzed 1610 subjects who passed all three attention checks (April = 810, May = 800).

Justification of sample size for both surveys was pre-registered through the Open Science Framework (prewave: https://osf.io/z9yj3/; first-wave: https://osf.io/jh7gk/). Sample size for the prewave survey was guided by a power analysis, where we sought to detect a Cohen's $d$ = .25 with 80% power in a between-subjects experimental manipulation run after participants completed the prewave survey (manipulation not reported here). This analysis guided the sample size in the first-wave survey, which included no experimental manipulation.

### Materials and design

We developed nine scales, each composed of three to five items, to measure beliefs about (1) the health risks and social distancing requirements about COVID-19 (i.e., perceptions of the severity of the illness and the need to prevent its spread), (2) forms of scientific and medical distrust (e.g., antivaccination attitudes, degree of trust in large medical organizations, attitudes towards a then-hypothetical COVID-19 vaccine), and (3) politically-motivated considerations pertaining to the pandemic response (e.g., responses to the pandemic made by politicians and members of the media). Seven of these scales were included in the prewave survey and eight in the first-wave survey. In the prewave survey one item was removed from the COVID-19 vaccine fears scale to increase internal reliability of the scale; we improved this item for the first-

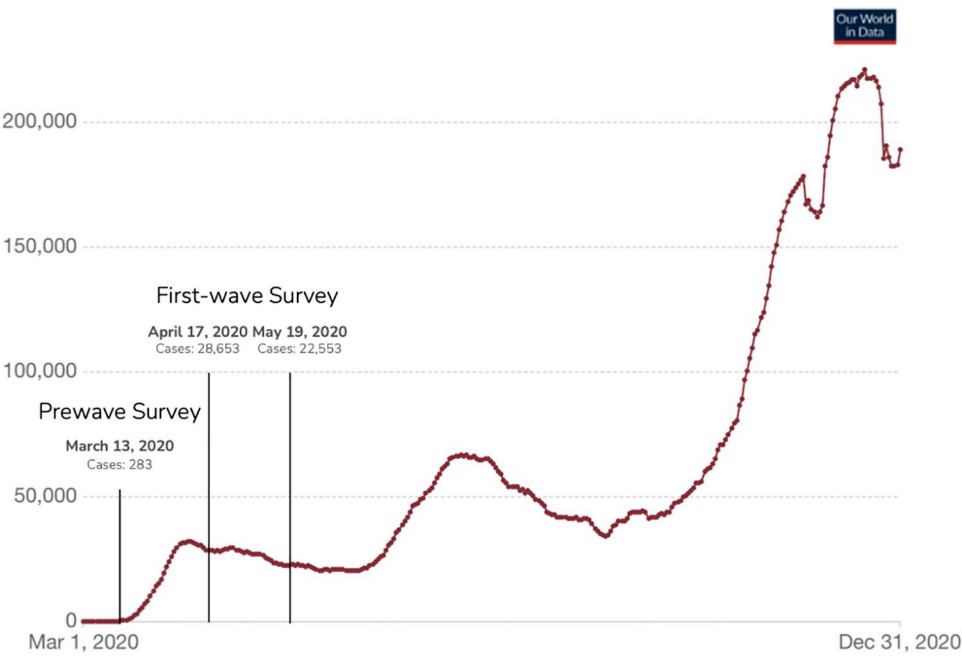

**Fig 1. Dates on which surveys were administered in relation to rolling 7-day average of reported COVID-19 cases in the U.S during 2020.** Visualization of cases was produced using our world in data COVID-19 data tracker at ourworldindata.org. Case data provided by the COVID-19 data repository at the Center for Systems Science and Engineering (CSSE) at Johns Hopkins University.

wave survey. The media distrust scale was only included in the prewave survey. As the pandemic developed, two additional scales were added for the first-wave survey: attitudes about opening the economy, and attitudes about general vaccine efficacy (see Table 1 for example questions). Scales were preregistered and are provided on the OSF links given above (also see

**Table 1. Examples of questions on nine scales designed to assess attitudes related to COVID-19.**

| Scale Name | Example Item | Survey |
|---|---|---|
| *Coronavirus Severity* | COVID-19, commonly referred to as coronavirus, is no more severe than the flu. | prewave, first-wave |
| *Distrust Democratic Politicians* | Some politicians are making a big deal out of COVID-19 for political gain. | pre, first |
| *Anti-COVID-19 Vaccine Attitudes* | I fear the government will use COVID-19 as an excuse to mandate vaccinations. | pre, first |
| *Medical Organization Distrust* | Medical organizations like the CDC and WHO are untrustworthy. | pre, first |
| *Media Distrust* | Some reporters and members of the media are making COVID-19 seem like a bigger deal than it really is. | pre only |
| *Origins of COVID-19* | COVID-19 was engineered in a laboratory. | pre, first |
| *Foreign Threat* | One of the best ways to reduce the spread of COVID-19 is to stop immigration into the United States. | pre, first |
| *Vaccine Effectiveness* | Your chances of getting a disease after being vaccinated against it are incredibly low. | first only |
| *Opening the Economy* | We should stop social distancing as soon as possible to kickstart the economy. | first only |

Supplemental Online Materials). These scales were collectively intended to assess how an interconnected web of beliefs and attitudes shaped Americans' early perceptions of COVID-19.

## Procedure

Prior to beginning the survey, participants were informed that it was designed to understand the public's perception of COVID-19. Participants first read a one-paragraph description of COVID-19 and rated how familiar they were with the virus on a seven-point Likert scale (from "Very Unfamiliar" to "Very familiar"). Next, participants responded to a series of questions designed to assess their beliefs and attitudes about COVID-19 and related issues (see Table 1 above). Participants responded to items on a seven-point Likert scale with values ranging from "Strongly Disagree" to "Strongly Agree". They first responded to items asking about their general perception of the severity of COVID-19 and its risks, and then proceeded to items for the other measures. After completing the prewave survey, participants performed an experimental task (not reported here).

All items for a given scale appeared consecutively. Scales were divided into two higher-order groupings. Those relating to scientific beliefs (e.g., scales relating to vaccine safety and medical institutions) were presented in sequence, as were those relating to other considerations (e.g., scales relating to foreign threats associated with the illness, and distrust in politicians communicating the risks of the illness). The order of these two higher-level groupings was randomized across participants. Before completing the study, participants provided demographic information, including region of the U.S. in which they live, their age and identified gender, and their political identification regarding social and fiscal issues.

## Results

### Prewave attitudes

To identify which beliefs shaped COVID-19 skepticism before the first wave of cases struck the U.S., we predicted COVID-19 skepticism as a function of all belief predictors measured in the prewave study and the participant's political stance towards social issues. Because the predictors are strongly correlated (by design, as correlations between predictors is necessary to measure coherence), a high degree of collinearity between predictors may hinder interpretation of the findings. In Bayesian linear models, collinearity between predictors results in flat (uninformative) posteriors estimates, even if one of the predictors in fact is predictive [37]. But as will be seen, this is not the case in our full model (reported below), and hence is not a concern for these analyses.

Survey data and R scripts for reproducing the reported analyses and figures are available with open access at the Open Science Framework: https://osf.io/8xerq/?view_only=df620c3e86984668ae1a225028b5cd9b. For details of data analyses, see Supplemental Online Materials.

For the prewave survey, Fig 2 shows the effect for each of the predictors in the model on COVID-19 severity, where more negative values imply the belief had a stronger negative effect on COVID-19 severity judgments (i.e., increased COVID-19 skepticism). Distrust in the intentions of Democratic politicians and fear of a COVID-19 vaccine were the only two reliable predictors of taking the virus less seriously; foreign threat assessment was the only predictor of taking the virus more seriously.

Fig 3 shows the conditional effects of the three most reliable predictors of responses on the COVID-19 severity scale. These are conditional effects (i.e., the relationship between the two displayed variables is conditioned on the mean value for the remaining scales). Two of the

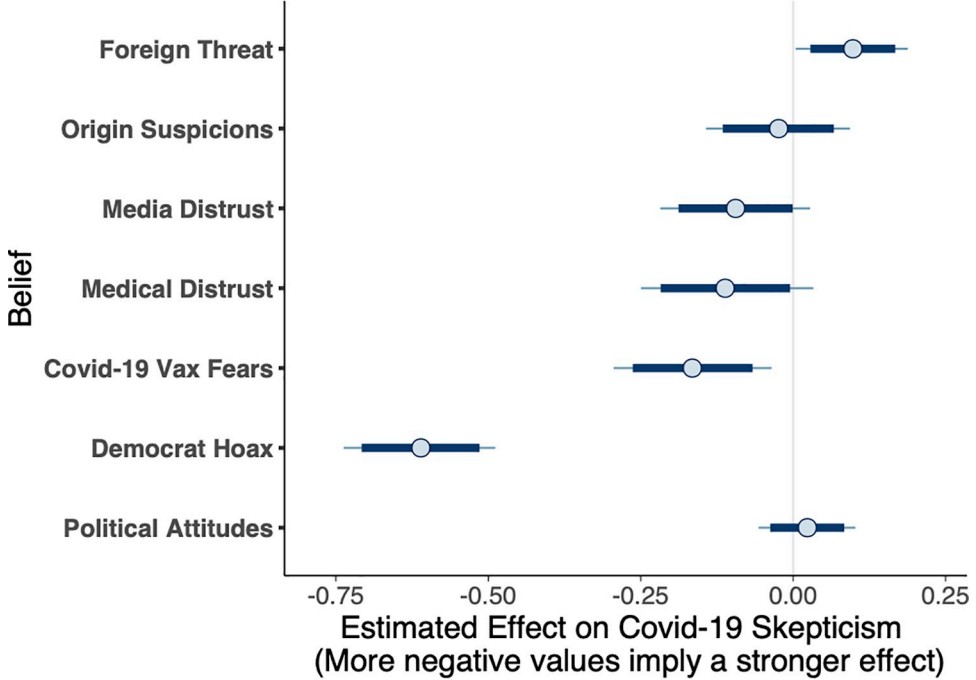

**Fig 2. Posterior regression estimates for predictors of prewave COVID-19 attitudes.** Circle = mean, thick bar = interior 95%, thin bar = interior 99% of posterior distribution.

predictors, distrust of Democratic politicians (Fig 3A) and COVID-19 vaccine fears (Fig 3B), predicted taking the virus less seriously; while one of the predictors, foreign threat (Fig 3C), predicted taking the virus more seriously. In addition to the model estimates, Fig 3 displays each participant's averaged response on the scale, colored by political stance. This jittered data, along with the kernel density plots in the margins of the panels (a-c), highlight the complex relationship between political ideology and COVID-19 skepticism. The three density plots

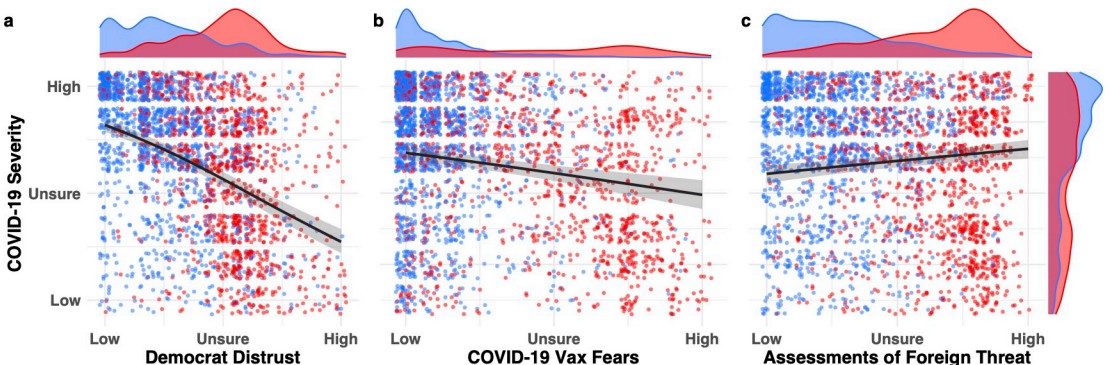

**Fig 3. Histograms of three strongest predictors of prewave COVID-19 skepticism and their relation to political ideology.** Regression lines represent effect size estimates conditioned on the average response for the remaining predictors in the maximal model. Error regions represent 95% credible intervals (the interior 95% of the posterior distribution for the effect size). Participant-level averaged responses on the scales for those who identified as socially liberal (blue) and conservative (red) are jittered with kernel density plots representing the distribution of responses for each predictor. Responses of participants who identified as moderate were excluded from visualization but were included in the model. Horizontal density functions at top of each panel summarize distribution of responses by Democrats and Republicans on each predictor variable; vertical density functions at right summarize distribution of responses on the COVID-19 severity scale.

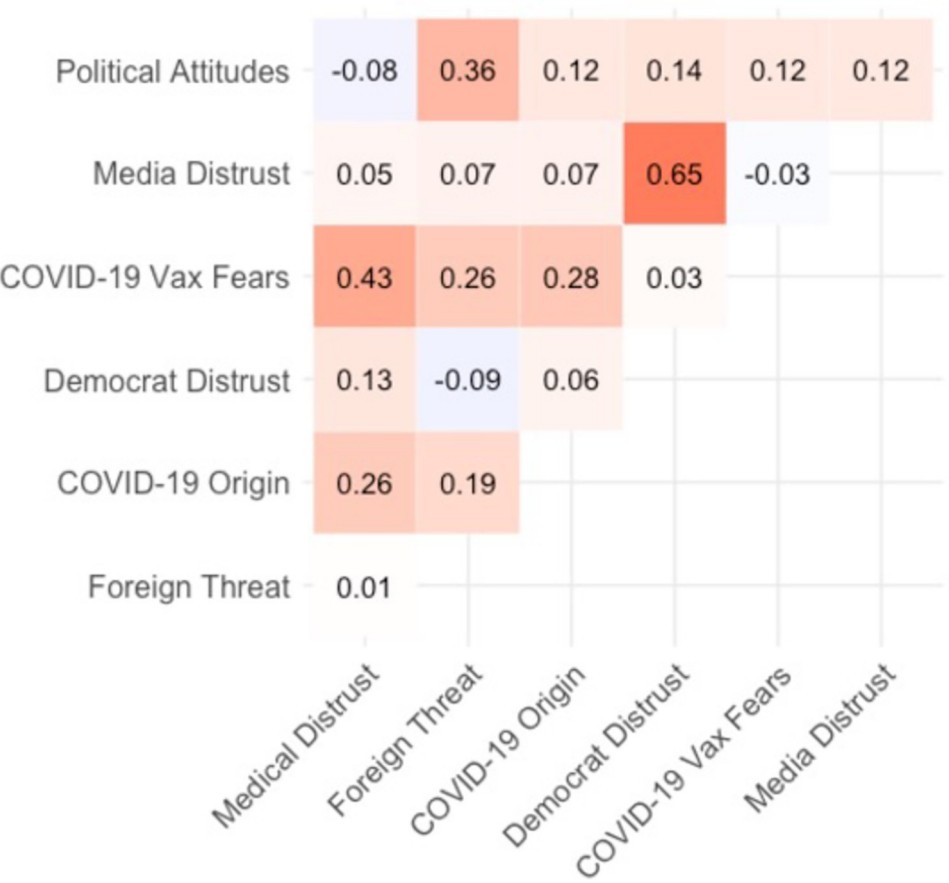

**Fig 4. Pairwise partial correlations between beliefs based on prewave survey.** Higher values on the Political Attitudes scale implies more conservative political stance. The shading on each cell represents correlation value (darker red = stronger positive; darker purple = stronger negative).

running horizontally at the top of each of the three panels show the degree of separation between how Democrats (blue) and Republicans (red) responded on the three measures, with Democrat responses being skewed to the right and Republican responses to the left. Note that political polarization (the degree of separation between Democrats and Republicans) is far larger for the three predictors than for the COVID-19 severity scale itself: the separation between the horizontal blue and red density functions at the top of each of the three panels is much greater than the corresponding separation for the vertical density functions on the right of the figure (i.e., for the COVID-19 severity scale).

In fact, posterior estimates of the regression coefficients from the maximal model revealed that after other beliefs were accounted for, there was no direct effect of political affiliation on COVID-19 skepticism ($b = 0.02$, 95% CI [-0.04, 0.08]). This finding indicates that beliefs other than direct political attitudes were the most effective predictors of prewave COVID-19 skepticism. Foreign threat assessment (believing that the U.S. should halt all immigration, coupled with xenophobic attitudes, such as that contact with Chinese people should be avoided to reduce the risk of contracting the virus) was the sole predictor of taking the virus more seriously ($b = .10$, 95% CI [0.03, 0.17]), and was the belief most reliably correlated with conservative attitudes ($r_{partial} = .36$). Pairwise partial correlations between all the predictors measured in the survey are shown in Fig 4. The stark political polarization surrounding foreign threat

calculations, and its positive effect on rated COVID-19 severity, illuminates how political attitudes shaped COVID-19 attitudes indirectly through auxiliary beliefs.

The strongest predictor of undervaluing the health risks associated with COVID-19 was distrusting the intentions of Democratic politicians (e.g., believing Democrats exaggerated the COVID-19 health risks for political gain; $b$ = -0.61, 95% CI [-0.71, -0.51]). Distrust of the media's coverage of the virus was moderately correlated with distrust of Democratic politicians ($r_{partial}$ = .65); however, media distrust only weakly predicted COVID-19 severity ($b$ = -0.09, 95% CI [-0.19, 0.00]). As these two predictors were uniquely correlated with one another, it suggests that these two beliefs represent a more general ontology of political distrust, which served to undermine propensity to consider the pandemic to be a serious public health threat prior to the onslaught of cases.

The second strongest predictor of COVID-19 skepticism was fearing future COVID-19 vaccination requirements ($b$ = -0.17, 95% CI [-0.26, -0.07]). Even though vaccines had not yet been developed (indeed, COVID-19 had not yet impacted the U.S.), pre-seeded fears of the government requiring vaccination played a role in sowing early distrust about the virus. These fears moderately cohered with distrust of large medical organizations ($r_{partial}$ = .43), which was moderately predictive of COVID-19 skepticism ($b$ = -0.11, 95% CI [-0.22, -0.01]). Vaccine fear also cohered with concerns about the origins of the virus ($r_{partial}$ = .28), which was not predictive in the full model ($b$ = -0.02, 95% CI [-0.12, 0.07]), and with evaluations of foreign threat ($r_{partial}$ = .26). Given that distrust of large medical organizations and having concerns about the origins of the virus were also weakly correlated ($r_{partial}$ = .26), this overall pattern suggests these three beliefs formed a tightly connected triad acting as an ontology of medical science distrust, which may have operated independently of political distrust in shaping COVID-19 skepticism.

Two sources of evidence thus suggest that two separable ontologies of distrust independently shaped skepticism before the first wave of cases. First, partial correlations between the predictors reveal two clusters of predictors––distrust in Democratic politicians and media, versus COVID-19 vaccine fears, distrust in medical organizations, and beliefs about the origin of the virus. Each cluster is internally correlated, but the two clusters are relatively independent of one another. Second, comparing the leave-one-out (loo) cross-validation accuracy of the full model with subsets of that model reveals that the most parsimonious, best-predicting model includes only distrust of Democrats and COVID-19 vaccine fears as predictors of COVID-19 skepticism. As shown in Fig 5, which shows the loo cross-validation accuracy of the full model and multiple subset models, the model predicting COVID-19 severity as a function of distrust in democratic politicians and COVID-19 vaccine fears while excluding political attitudes ("Big 2—politics") is the model with the smallest set of predictors with a predictive accuracy within the standard error of the most predictive model (the maximal model with all the predictors). Furthermore, the model that only includes participants' political attitudes is by far the worst-fitting model. These results suggest that auxiliary beliefs, rather than political polarization *per se*, are critical in predicting COVID-19 skepticism.

## First-wave attitudes

As depicted in Fig 6, health risks associated with COVID-19 was taken more seriously during the first wave than before the first wave ($b$ = 0.75, 95% CI [0.43, 1.06]). The magnitude of this shift was roughly equal across the political spectrum ($b$ = -0.04, 95% CI [-0.12,.04]), with more conservative participants giving lower ratings of severity at each time point ($b$ = -0.34, 95% CI [-0.38, -0.30]). However, the magnitude of shift was slightly larger among the most socially liberal participants than among the most socially conservative. This increase in rated severity

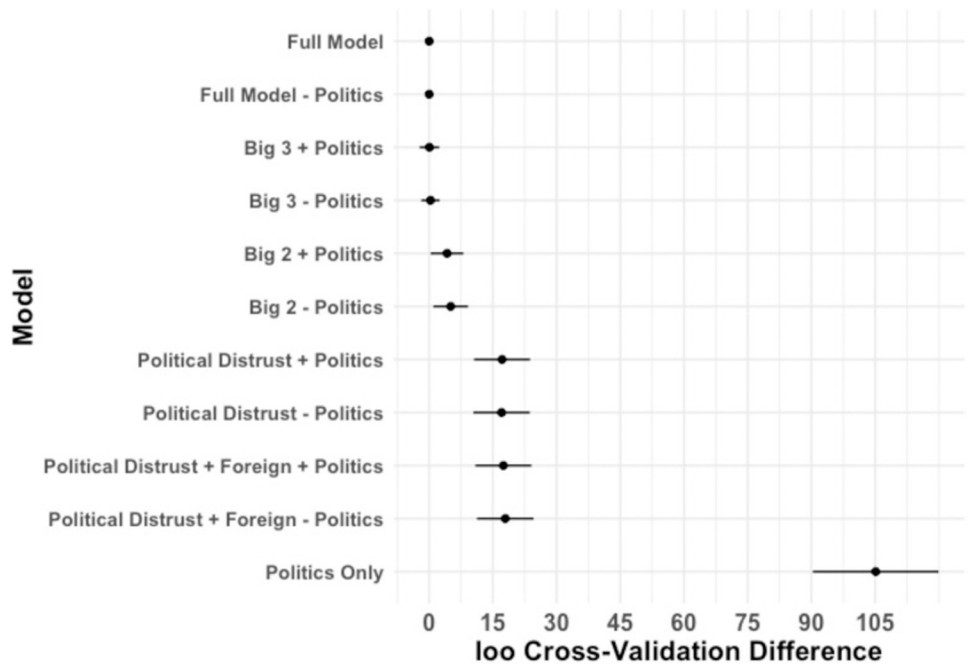

**Fig 5. Model comparisons predicting COVID-19 attitudes for the prewave survey using leave-one-out (loo) cross-validation information criteria.** Error bars indicate SE of estimate. The simplest model within the SE of the full model, "Big 2 –politics", includes the two strongest predictors of skepticism (distrusting Democratic politicians and COVID-19 vaccine fears), and does not include participants' political attitudes.

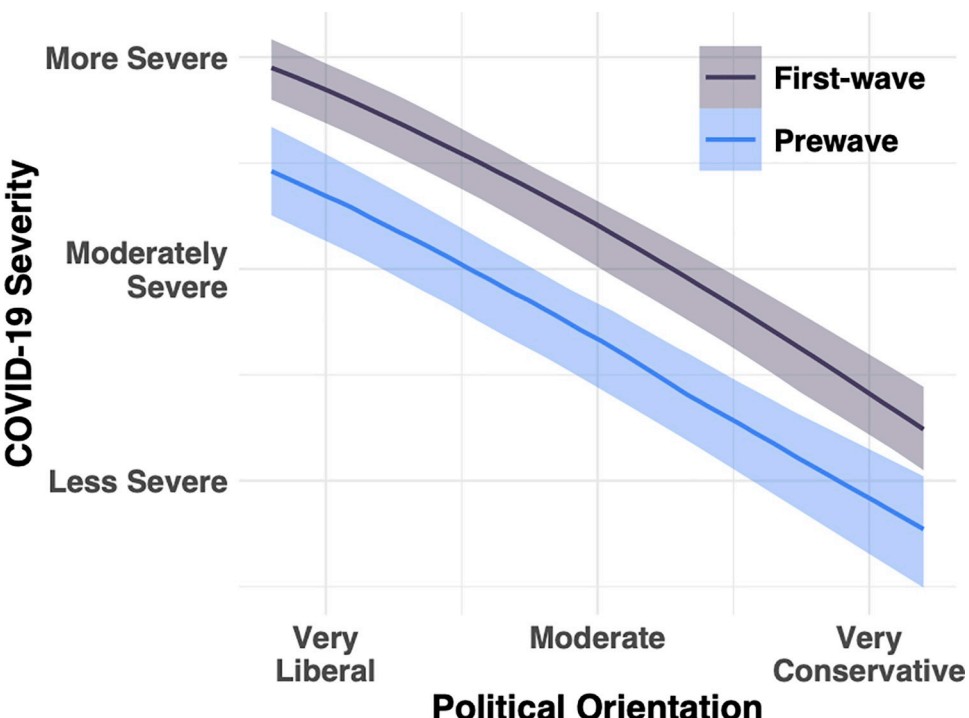

**Fig 6. Shift in severity ratings from prewave to first-wave surveys as a function of political positions.** Error bars = 95% credible intervals of effect of political beliefs on COVID-19 severity.

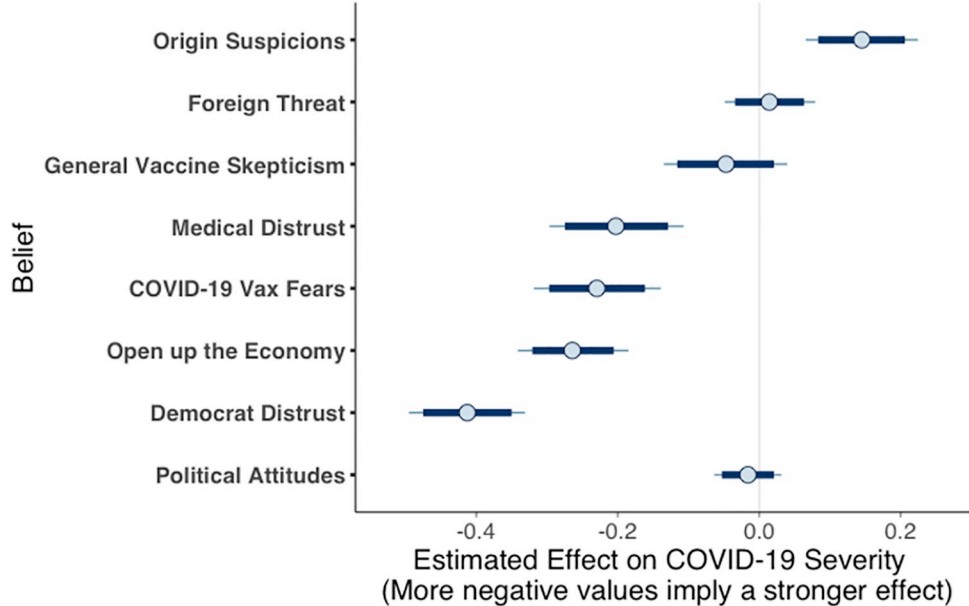

**Fig 7. Posterior regression estimates for predictors of first-wave COVID-19 attitudes.**

suggests that the onset of cases in the United States (more than 20,000 cases per day during the first wave) had a measurable, albeit small, effect on people's attitudes across the entire political spectrum.

To provide a more detailed analysis of the factors that impacted COVID-19 skepticism on the first-wave survey, we examined a model including the mean response on each belief scale and the participant's political attitudes towards social issues. As shown in Fig 7, this full model showed that (as was the case for prewave attitudes), political position was not predictive of COVID-19 severity judgments once other predictors were accounted for ($b$ = -0.02, 95% CI [-0.05, 0.02]). Whereas prewave severity judgments had one strong (Democrat political distrust) and one moderate (COVID-19 vaccine fears) predictor, four strong predictors emerged in the first-wave survey: (1) distrust of Democratic politicians ($b$ = -0.41, 95% CI [-0.47, -0.35]); (2) desire to reopen the economy ($b$ = -0.26, 95% CI [-0.32, -0.21]); (3) COVID-19 vaccine fears ($b$ = -0.23, 95% CI [-0.30, -0.16]); and (4) distrust of large medical organizations ($b$ = -0.20, 95% CI [-0.27, -0.13]).

One additional belief weakly predicted taking the virus more seriously: uncertainty about the (lab versus natural) origins of the virus ($b$ = 0.14, 95% CI [0.08, 0.21]). During the first wave, origin attitudes played a similar role as foreign threat calculations did in the prewave survey; the latter variable was not a reliable predictor during the first wave ($b$ = 0.00, 95% CI [-0.05, 0.05]). In the first-wave survey, those favoring conservative ideology were more likely to believe the virus originated in a lab, suggesting a complex relationship between political ideology, scientific distrust, and COVID-19 skepticism. General vaccine skepticism has no effect on COVID-19 skepticism during the first wave ($b$ = -0.05, 95% CI [-0.12, 0.02]), suggesting that concrete fears about a COVID-19 vaccine (such as mandatory vaccine requirements) were more consequential in determining people's attitudes than were general beliefs about vaccine efficacy.

Fig 8 shows the effects of the five strongest predictors of COVID-19 attitudes in the first wave. Four of these beliefs predicted taking the virus less seriously, whereas one belief predicted taking the virus more seriously. In addition to the model estimates, Fig 8 displays each

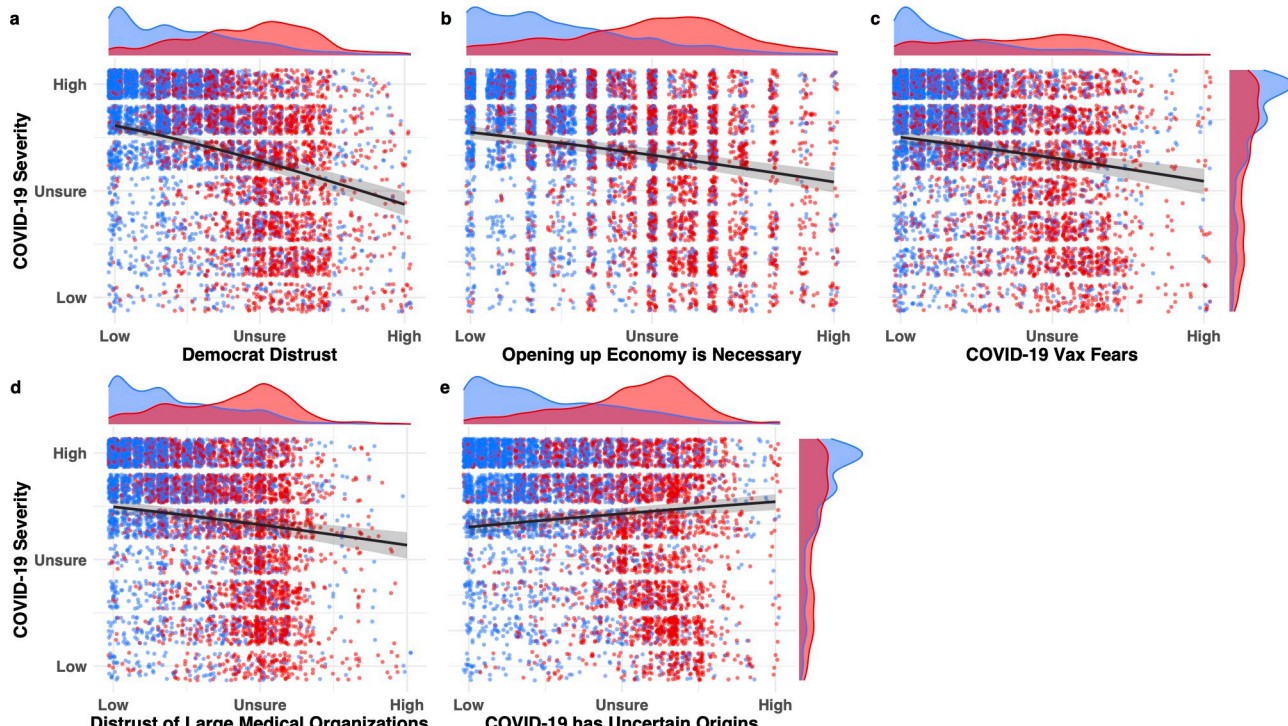

**Fig 8. Histograms of five strongest predictors of first-wave COVID-19 skepticism and their relation to political ideology.** Regression lines represent effect size estimates conditioned on the average response for the remaining predictors in the maximal model. Error regions represent 95% credible intervals (the interior 95% of the posterior distribution for the effect size). Participant-level averaged responses on the scales for those who identified as socially liberal (blue) and conservative (red) are jittered, with kernel density plots representing the distribution of responses for each predictor. Responses of participants who identified as moderate were excluded from visualization but were included in the model. Scale labels are recoded for readability from the original 7-point Likert scale (Strongly Disagree = Low, Neither Agree nor Disagree = Unsure), Strongly Agree = High). Horizontal density functions at top of each panel summarize distribution of responses by Democrats and Republicans on each predictor variable; vertical density functions at right summarize distribution of responses on the COVID-19 severity scale.

participant's averaged response on the scale colored by political stance. The five density plots running horizontally at the top of each of the five panels (a-e) show the degree of separation between how Democrats (blue) and Republicans (red) responded on the five measures, with Democrat responses being skewed to the right and Republican responses to the left. As was the case for the prewave survey (Fig 3), each of the predictor variables shows greater separation between Democrats and Republicans than did responses on the COVID-19 severity scale itself. (Note greater separation of horizontal blue and red density functions at top of each panel compared to the separation of vertical blue and red density functions at right of figure.) Thus, the degree of polarization in attitudes across partisan lines was *greater* for beliefs about the predictor variables than for beliefs directly about the severity of COVID-19.

Leave-one-out cross-validation provided further evidence that by the first wave, COVID-19 attitudes had become more deeply intertwined with a wider array of considerations. Fig 9 shows the loo cross-validation performance of the full model and models that include subsets of the predictors. The model with the smallest set of predictors (the most parsimonious model) within the standard error of the predictive accuracy of the best performing model included the five most reliable predictors of COVID-19 attitudes but not political affiliation ("Big 5 –politics"). Four of these beliefs predicted being skeptical in the severity of the virus (Democratic distrust, opening the economy, COVID-19 vax fears, and general medical distrust) and one predicted taking the virus more seriously (origin concerns). With four of these predictors

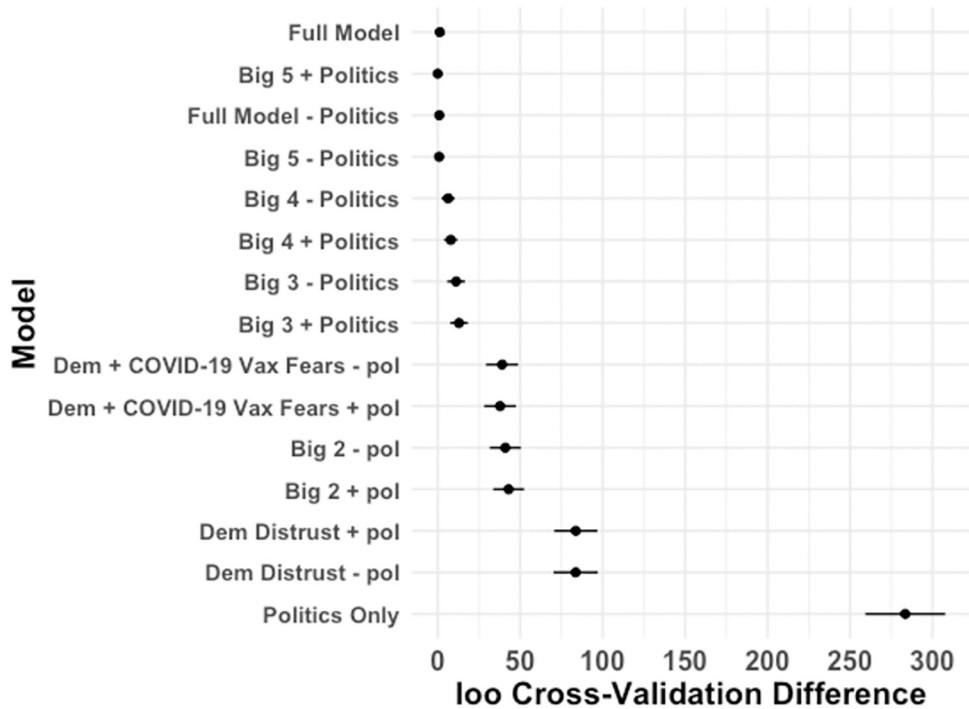

**Fig 9. Model comparisons predicting COVID-19 attitudes for the first-wave survey using leave-one-out (loo) cross-validation information criteria.** Error bars indicate SE of estimate. The simplest model within the SE of the full model, "Big 5 –politics", includes the five strongest predictors of responses on the COVID-19 severity scale (i.e., the five predictors shown in Fig 8), and does not include participants' political attitudes.

implying taking the virus less seriously, skeptical COVID-19 attitudes appeared to be strongly embedded within a larger coherent network of beliefs, as compared to prewave attitudes for which only two beliefs predicted more skeptical attitudes (one strongly and one moderately). This pattern suggests that a coherence shift altered attitudes during the first wave of cases.

In prewave attitudes, we found that two preexisting ontologies of distrust shaped early COVID-19 attitudes. We investigated whether the same latent structures were present in predictors of first-wave attitudes. Fig 10, which shows the pairwise partial correlations among the full set of predictors, suggests how beliefs were interconnected. The two strongest predictors, opening the economy and distrusting Democratic politicians, had a moderate correlation with one another ($r_{partial}$ = .45). As these beliefs are about non-scientific content, this strong correlation (coupled with the pronounced relationship of each variable with COVID-19 attitudes) suggests that considerations beyond science and immediate health risks drove skepticism in laypeople's reasoning about COVID-19 from the prewave into the first wave. Like prewave attitudes, a separate cluster of science-oriented beliefs shaped COVID-19 skepticism on the first-wave survey. The impact of medical skepticism was stronger during the first wave, though it was a weaker predictor relative to non-science beliefs about politicians and the economy.

As also shown in Fig 10, COVID-19 vaccine fears and distrust of large medical organizations were correlated ($r_{partial}$ = .25), and general vaccine skepticism was moderately correlated with COVID-19 vaccine fears ($r_{partial}$ = .49). This pattern suggests that COVID-19 vaccine fears, which correlated with both distrust in medical organizations and general skepticism about the efficacy of vaccines, likely served as the central node in the cluster of beliefs expressing distrust of medical science. It appears that that preexisting ontologies of distrust identified in the prewave survey—one centering around political considerations and the other around

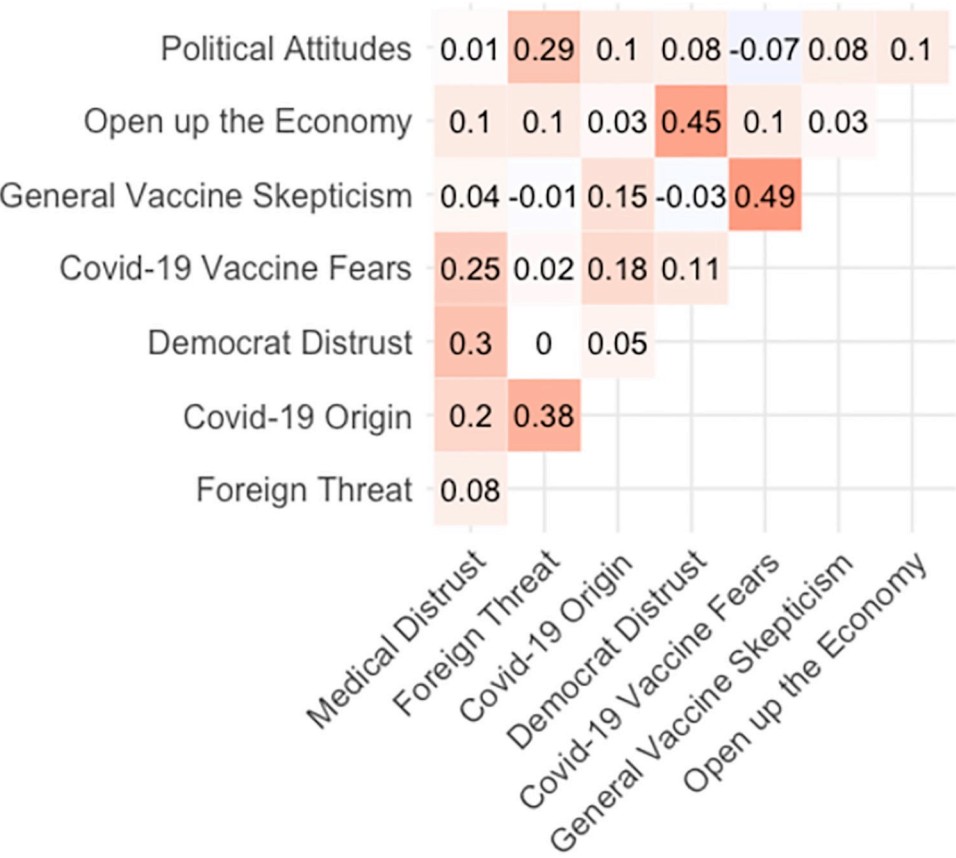

**Fig 10. Pairwise partial correlations between beliefs based on first-wave survey.** Higher values on the political attitudes scale implies more conservative political stance. The shading on each cell represents correlation value (darker red = stronger positive; darker purple = stronger negative).

medical science—persisted through the first-wave. However, the number of predictors expressing these two ontologies doubled from prewave to first-wave, suggesting that the influence of these latent ontologies on COVID-19 attitudes grew as the pandemic progressed. More notably, beliefs representing medical science distrust increased in their influence, suggesting that skeptical attitudes became more coherent with distrusting views of medical science. As in the prewave survey, the variable most strongly correlated with conservative politics was assessment of foreign threat ($r_{partial}$ = .29), which was in turn most reliably correlated with COVID-19 origin ($r_{partial}$ = .38).

## Discussion

Our analyses of attitudes related to COVID-19, based on surveys administered in the spring of 2020 just prior to and then during the first-wave of COVID-19 cases in the United States, offer insight into the origins of widespread skepticism about the severity of the virus threat, and eventually about cost-benefit tradeoffs for a vaccine. Consistent with other sources of evidence [7–11], beliefs about COVID-19 at its onset were sharply polarized by political leaning, with conservatives tending to minimize threat severity relative to liberals. However, detailed analyses revealed that the central factors driving this divergence were two preexisting ontologies of skepticism about sources of information: distrust of Democratic politicians, and distrust of medical science. In neither survey was political leaning *per se* a reliable predictor of

assessments of COVID-19 severity after accounting for the influence of these two types of skepticism. Hypothetically, a committed Republican need not necessarily believe that Democratic politicians and medical scientists are liars and might well take COVID-19 seriously.

It is also notable that belief in a foreign threat and/or the possible origin of the virus in a Chinese lab—beliefs more likely to be held by conservatives—were associated with *increased* evaluation of COVID-19 severity. Thus, conservatives overall were more skeptical of COVID-19 severity due to their acceptance of the two ontologies that supported such skepticism, *despite* the impact of a countervailing factor. Rather than political ideology *per se* being a monolithic and central predictor of beliefs about the virus, it appears to be more accurately construed as a correlate of a broader pattern of ancillary beliefs that shaped COVID-19 attitudes.

One striking finding of our study is that a key predictor of skepticism was fear of an eventual vaccine to treat the virus. This apprehension was reflected in agreement with statements such as, "I fear the government will use COVID-19 as an excuse to mandate vaccinations." Vaccine fear was a major predictor of lower judged COVID-19 severity even in the prewave survey of March 2020—when the disease had minimally impacted the United States, and about nine months before any vaccine became available to the public. By the time vaccines became a reality, vaccine fear had merged with other politically-tinged attitudes that led to widespread resistance to efforts to encourage vaccination, particularly among Republicans [12]. By summer 2021, a resurgence of COVID-19 among unvaccinated individuals led many business and political leaders to shift from encouraging to mandating vaccination. Thus, the early vaccine fear that was partly responsible for minimizing the perceived threat of COVID-19 would ultimately discourage adoption of the most effective means to end the pandemic, thereby provoking mandates of the sort that had been feared in the first place.

It is also notable that the preexisting ontologies of skepticism—distrust of Democratic politicians and of medical science—broadened and strengthened from the prewave to the first-wave survey. The number of beliefs impacting skepticism about COVID-19 increased, suggesting that preexisting attitudes had become entwined in a wider web of considerations, such as more generalized distrust in medical science. It seems that people's belief systems underwent a coherence shift [16–18] as the pandemic progressed. As skeptics solidified their attitude towards the virus, they more strongly endorsed additional beliefs that supported their skepticism. Based on our analyses of the cognitive mechanism underlying COVID-19 skepticism, we anticipate that a coherence-generating mechanism will also be at play in shaping attitudes towards future societal threats. Given the evidence that as the pandemic progressed, anti-science attitudes became more deeply embedded within a network of skeptical beliefs, it is crucial that gatekeepers of scientific evidence are *early* and *active* in dissociating the results of scientific methodology from previously-held attitudes towards political bodies and organizations. As time progresses, attitudes stabilize and calcify. Therefore, when informing the public of future medical threats, medical organizations such as the Centers for Diseases Control and World Health Organization should prepare additional educational information accompanying the scientific research, designed to counter connections between preexisting forms of distrust (often political) and beliefs about incoming scientific information.

The emergence and maintenance of COVID-19 skepticism appears to be similar in form to climate-change denialism [38]. Both misconceptions are based on conspiracy theories that posit malevolent intentions of political and scientific organizations [24,28,39]. Conspiracy theories promote narratives alleging that academic and scientific organizations are conspiring with Democratic politicians and the mainstream media to push political agendas under the guise of scientific rigor [27]. For example, skeptics of climate change often claim that Democratic politicians are conspiring with climate scientists to use scientific evidence to regulate the

economy for liberal (or socialist) ends [40]. In the case of COVID-19 denialism, conspiracy theories have claimed that that the virus was spread on purpose, and that medical professionals and Democratic politicians were conspiring to make then-President Donald Trump look bad in an election year [41].

Once people become skeptical of countervailing sources of information, the internet will feed them information that confirms their beliefs [42], yielding dramatic belief polarization [43]. During the pandemic, conspiracy theories and misinformation circulated within more conservative circles on the internet [44], sowing distrust of established scientific facts about the virus [45] and agents promoting it [41,46,47]. In general, misinformation did not promote anti-science attitudes by sowing distrust in the scientific method, but rather by sowing distrust in scientists and those who follow their advice [48]. Indeed, some of the most prominent spreaders of COVID-19 misinformation were a handful of medical doctors acting as internet "influencers" [35], who claimed to have "inside knowledge" of medical conspiracies. Thus, anti-science attitudes are largely the product of conspiracy theories, rather than of inaccurate scientific or conceptual knowledge [15,24].

The seeds of skepticism about the severity of COVID-19, and about the desirability of a vaccine, can be found in preexisting skepticism about the credibility of medical science and its proponents. Once experts and "impartial" sources of information have been discredited by attacks on their motives, the public will no longer share a common base of facts to guide opinion about key social issues. Educational interventions that promote rational evaluation of scientific evidence [49] or rational cost-benefit analysis [50] can have some impact on people's decisions, but persistent distrust of the generators of knowledge will work to sustain anti-science views even in the face of counterevidence. It will likely be necessary to confront such distrust head on to bring about positive change in how the public responds to present and future societal threats.

## Supporting information

**S1 File.**
(PDF)

## Author Contributions

**Conceptualization:** J. Hunter Priniski, Keith J. Holyoak.

**Data curation:** J. Hunter Priniski.

**Funding acquisition:** Keith J. Holyoak.

**Methodology:** J. Hunter Priniski.

**Validation:** J. Hunter Priniski.

**Visualization:** J. Hunter Priniski.

**Writing – original draft:** J. Hunter Priniski.

**Writing – review & editing:** J. Hunter Priniski, Keith J. Holyoak.

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
