## [Decision Letter · Decision Letter 0]

22 Dec 2021

PONE-D-21-28659A Darkening Spring: How Preexisting Distrust Shaped COVID-19 SkepticismPLOS ONE

Dear Dr. PRINISKI,

Thank you for submitting your manuscript to PLOS ONE. After careful consideration, we feel that it has merit but does not fully meet PLOS ONE’s publication criteria as it currently stands. Therefore, we invite you to submit a revised version of the manuscript that addresses the points raised during the review process. Please revise the paper according to the reviewers' comments. Please try to provide more insight on the methodology used in the paper.

We look forward to receiving your revised manuscript.

Kind regards,

Camelia Delcea

Academic Editor

PLOS ONE

Journal Requirements:

Preparation of this paper was supported by NSF Grant BCS-1827374.

Preparation of this paper was supported by NSF Grant BCS-1827374.

Preparation of this paper was supported by NSF Grant BCS-1827374.

Reviewers' comments:

Reviewer's Responses to Questions

**Comments to the Author**

1. Is the manuscript technically sound, and do the data support the conclusions?

Reviewer #1: Yes

Reviewer #2: No

Reviewer #3: Yes

Reviewer #4: Yes

2. Has the statistical analysis been performed appropriately and rigorously? 

Reviewer #1: Yes

Reviewer #2: Yes

Reviewer #3: Yes

Reviewer #4: Yes

3. Have the authors made all data underlying the findings in their manuscript fully available?

Reviewer #1: Yes

Reviewer #2: No

Reviewer #3: Yes

Reviewer #4: Yes

4. Is the manuscript presented in an intelligible fashion and written in standard English?

Reviewer #1: Yes

Reviewer #2: Yes

Reviewer #3: Yes

Reviewer #4: Yes

5. Review Comments to the Author

Reviewer #1: The paper deals with a very important and actual topic. There are just a few comments/suggestions that I would like to make:

1. clarify the number of scales used in pre-wave (6 in text, 7 in table)

2. recheck the numbering of figures and their reference in text (figure 1 appears twice)

3. recheck the interpretation of the skewness (left, right) for democrats vs. republicans for all distributions

4. check for multicollinearity prior to presenting CI

5. recheck the interpretation of correlation coefficients (r=.43 is moderate, not strong)

6. consistent use of the same variable names in each figure/table/ text (eg. democrat distrust and democrat hoax, political attitude and conservative politics). If they refer to different variables, please explain the differences.

Overall, this paper can be viewed as a good presentation of the insights of the skeptical attitude regarding covid.

Reviewer #2: The contribution of the current paper is not sufficient for publication in this journal . Thus, I have decided to reject the paper "A Darkening Spring: How Preexisting Distrust Shaped COVID-19 Skepticism".

Reviewer #3: Solid work. Would have liked additional comments on the development of the framework and specifics to address potential new narratives as to inform better policy that takes into account the insights provided for future issues.

Reviewer #4: Your manuscript was well thought out, clear and provides important information that helps explains the current beliefs about COVID. The introduction provided a good context on the topic. The method on what was done was clear and well designed. The results showed what attitudes were related and how that led to current belief systems about COVID. The figures and tables added a needed component to help explain what was discovered. The conclusion was appropriate based on the results. I recommend your paper for publication.

6. PLOS authors have the option to publish the peer review history of their article (what does this mean?). If published, this will include your full peer review and any attached files.

Reviewer #1: No

Reviewer #2: No

Reviewer #3: **Yes: **José Bayoán Santiago Calderón

Reviewer #4: **Yes: **Jill A. Yamashita

---

## [Author Response · Author response to Decision Letter 0]

12 Jan 2022

Response to Reviewers

Hunter Priniski Keith Holyoak

We thank the reviewers for their careful comments and address each point below. 

Journal Requirements

Completed

We added the following Ethics Statement subsection to our Methods section: 

Ethics Statement

 The Institutional Review Board of the University of California, Los Angeles, approved our study, and written consent was received for each participant. All participants were over 18 years of age. 

Preparation of this paper was supported by NSF Grant BCS-1827374.

Please state what role the funders took in the study: The funders had no role in study design, data collection and analysis, decision to publish, or preparation of the manuscript.

This info was added to our Acknowledgements section, which now reads: 

The funders of this study had no role in study design, data collection and analysis, decision to publish, or preparation of the manuscript.

Preparation of this paper was supported by NSF Grant BCS-1827374.

Preparation of this paper was supported by NSF Grant BCS-1827374.

Funding details have been removed: see revised Acknowledgement above.

Our data and analysis scripts are hosted publicly on the Open Science Framework. As one reviewer seems to have a tough time locating these files, we included a link to them in our Results section. 

We added an Ethics subsection to the Materials. See response to point 2 above. 

We did some small editing to the reference list for style. No references were added or removed. 

Reviewer Comments

Reviewer #1: 

We thank Reviewer #1 for their careful read of the manuscript. We individually address each point they raised below. 

1. clarify the number of scales used in pre-wave (6 in text, 7 in table)

We reported the wrong number in the original text. We updated the numbers, so they match. 

2. recheck the numbering of figures and their reference in text (figure 1 appears twice)

We updated the figure counts. 

3. recheck the interpretation of the skewness (left, right) for democrats vs. republicans for all distributions

We made these changes in the manuscript. And thank the reviewer for their careful eye. 

4. check for multicollinearity prior to presenting CI

The predictors will be strongly correlated by design as correlations between predictors is necessary to measure coherence. We thank the reviewer for raising concerns about the potential for a high degree of collinearity between the predictors which may undermine results. In Bayesian linear models, collinearity between predictors results in flat (uninformative) posteriors estimates, even if one of the predictors in fact is predictive. More information about Multicollinearity in posterior distributions of Bayesian multiple regression models can be found in R. McElreath’s textbook on Bayesian Modeling, Statistical Rethinking (pp. 141-142). 

As we saw in the full model, the posteriors of the predictors are not flat but rather informative, therefore alleviating concerns of collinearity between predictors. We added a justification for this in the manuscript before reporting the results. 

5. recheck the interpretation of correlation coefficients (r=.43 is moderate, not strong)

We updated our interpretations of the correlation coefficients such that they mirror conventional nomenclature. 

6. consistent use of the same variable names in each figure/table/ text (eg. democrat distrust and democrat hoax, political attitude and conservative politics). If they refer to different variables, please explain the differences.

The variable names were updated in both Figure 4 and 10, where we used conservative politics as a label for Political Attitudes. We changed both labels to Political Attitudes. We also fixed the labeling error in Figure 10 where we used Democratic Hoax rather than Democrat Distrust. This was fixed as well. 

Reviewer #2: no suggestions for revision

The contribution of the current paper is not sufficient for publication in this journal. Thus, I have decided to reject the paper "A Darkening Spring: How Preexisting Distrust Shaped COVID-19 Skepticism".

Reviewer #3: 

Solid work. Would have liked additional comments on the development of the framework and specifics to address potential new narratives as to inform better policy that takes into account the insights provided for future issues.

We added a suggestion in the discussion regarding how we believe a coherence mechanism will also be at play in future pandemics. Therefore, large medical organizations should be prompt in addressing these coherence-based connections between science and distrust. 

Reviewer #4: no suggestions for revision

---

## [Editor Report · Decision Letter 1]

14 Jan 2022

A Darkening Spring: How Preexisting Distrust Shaped COVID-19 Skepticism

PONE-D-21-28659R1

Dear Dr. PRINISKI,

We’re pleased to inform you that your manuscript has been judged scientifically suitable for publication and will be formally accepted for publication once it meets all outstanding technical requirements.

Kind regards,

Camelia Delcea

Academic Editor

PLOS ONE
---

## [Editor Report · Acceptance letter]

17 Jan 2022

PONE-D-21-28659R1 

A Darkening Spring: How Preexisting Distrust Shaped COVID-19 Skepticism 

Dear Dr. Priniski:

I'm pleased to inform you that your manuscript has been deemed suitable for publication in PLOS ONE. Congratulations! Your manuscript is now with our production department. 

Kind regards, 

on behalf of

Dr. Camelia Delcea 

Academic Editor

PLOS ONE